# The effect of mutations derived from mouse-adapted H3N2 seasonal influenza A virus to pathogenicity and host adaptation

Eun-Ji Choi[1][☯], Young Jae Lee[1][☯], Jin-Moo Lee[1], Yeon-Jung Kim[1], Jang-Hoon Choi[1], Byeongwoo Ahn[2], Kisoon Kim[1], Myung Guk Han[1]*

1 Division of Viral Disease Research, Center for Infectious Diseases Research, National Institute of Health, Korea Centers for Disease Control and Prevention, Osong, Republic of Korea, 2 College of Veterinary Medicine, Chungbuk National University, Cheongju, Republic of Korea

☯ These authors contributed equally to this work.

* mghan@korea.kr

**Data Availability Statement:** All relevant data are within the manuscript and its Supporting Information files.

## Abstract

Elucidating the genetic basis of influenza A viruses (IAVs) is important to understand which mutations will determine the virulence and the host range of mammals. Here, seasonal H3N2 influenza was adapted in mice by serial passage and four mutants, each carrying amino acid substitutions related to mouse adaptation in either the PB2, HA, NP, or NA protein, were generated. To confirm the contribution of each gene to enhanced pathogenicity and mouse adaptation, mice were inoculated with the respective variants, and virulence, replication, histopathology, and infectivity were examined. The virus harboring HA mutations displayed increased infection efficiency and replication competence, resulting in higher mortality in mice relative to those infected with wild-type virus. By contrast, the NP D34N mutation caused rapid and widespread infection in multiple organs without presenting virulent symptoms. Additionally, the PB2 F323L mutation presented delayed but elevated replication competence in the respiratory tract, whereas the S331R mutation in NA showed no considerable effects on mouse adaptation. These results suggested that mouse-adapted changes in HA are major factors in increased pathogenicity and that mutations in NP and PB2 also contribute to cross-species adaptability. Our findings offer a better understanding of the molecular basis for IAV pathogenicity and adaptation in a new host.

## Introduction

Seasonal influenza A viruses (IAVs) result in a substantial epidemic burden and are estimated to cause >645,000 annual respiratory deaths worldwide [1]. The recent outbreaks of H5N1 and H7N9 avian IAVs in humans have raised awareness regarding the emergence of new pandemic viruses [2, 3]. Most pandemic viruses gain the ability to infect and replicate in a new host either through adaptation or genetic reassortant [4, 5]. Moreover, amino acid substitutions in IAVs during adaptation in other hosts might be considered potential indicators of pre-pandemic viruses, which are associated with host range and pathogenicity [6–8].

**Funding:** This work was supported by grants of the National Institute of Health, Korea (2017-NI43001 and 2014-ER4301-02). The funder provided experiment resources, equipment, reagents and space, and helped to complete manuscript by pay the cost for English proofreading service, http://www.cdc.go.kr/index.es?sid=a5. The funder had no role in study design, data collection and analysis, decision to publish, or preparation of the manuscript.

**Competing interests:** The authors have declared that no competing interests exist.

Therefore, understanding the genetic markers responsible for host adaptation and virulence is important to improve the preparedness for the emergence of future pandemic viruses.

Mouse models have been used to elucidate adapted mutations related to host interactions and viral pathogenicity through experimental infection of mice with human IAVs [7, 9]. Generally, seasonal H3N2 viruses replicate poorly and display low virulence in mice. Therefore, infection and repeated passage are required for the adaptation of human IAVs to mice, resulting in mutations in multiple genes [7, 9]. In many cases, genetic mutations involved in mouse adaptation appear similar to those involved with human adaptation to avian viruses [7, 10, 11]. The IAV HA is a major factor in determining infectivity and virulence, and mouse-adapted substitutions in the HA gene frequently occur [9, 12–14]. Primarily, these mutations affect the preference of host receptor binding [10, 15] or are associated with loss of glycosylation sites [16, 17]. Genetic changes in the NP gene are rare during mouse adaptation; however, these mutations are considered important determinants of host range and mouse adaptation. The NP N319K mutation enhances interactions with importin α1, which increases nuclear import of ribonucleoprotein (RNP) [9, 18]. Additionally, the function of the NP D34N mutation is not yet known but might be related to interactions with PB2 [9]. Furthermore, the E627K and D701N substitutions in the PB2 gene have been commonly detected in mouse-adapted IAVs and related to enhanced virulence in mice [8, 9, 18–20]. These mutations are considered crucial markers of mammalian virulence and adaptation, suggesting PB2 as a major determinant of pathogenicity and host range. Although significant research has been conducted in these areas, the obvious genetic changes in IAVs that contribute to virulence and cross-species adaptability remain incompletely understood. Therefore, additional studies on the molecular basis of IAV pathogenicity and host adaptation are required.

Antigenic characterization of H3N2 virus is difficult to determine because of genetic diversification due to continual antigenic drift [21, 22]. The 2014–2015 H3N2 vaccine strain was antigenically mismatched to epidemic viruses and showed poor vaccine performance [23–25]. Therefore, the seasonal H3N2 virus, which showed genetically rapid evolution, has the potential to produce various mutations by adapting to other hosts [26].

Here, we infected mice with seasonal H3N2 IAV (A/Switzerland/9715293/2013) and obtained five mouse-adapted variants exhibiting enhanced pathogenicity. Sequence analysis identified a total of 20 amino acid substitutions in the PB2, PA, HA, NP, and NA genes, with most of these rare in circulating H3N2 IAVs. To confirm which mutations affect virulence and mouse adaptation, one of the five mouse-adapted viruses was chosen, and the exact role of each mutation in this virus was analyzed. We generated reassorted H3N2 viruses, each containing mouse-adapted mutations in either the PB2, HA, NP, or NA gene, and evaluated the mutations which contribute to mouse virulence and adaptation (e.g., based on morbidity, mortality, organ-specific replication, and lung histopathology). Our findings improve the understanding of the molecular basis for acquired pathogenicity and new host adaptation of IAVs.

## Materials and methods

### Animal and ethics statement

Six-week-old female BALB/c and DBA/1J mice were purchased from Orient Bio Inc. (Seoul, Korea). All animal experiments were carried out by trained researchers and approved by the Animal Experimental Ethics Committee of Korea National Institute of Health (approval No. KCDC-062-18-2A, KCDC-081-18-2A). According to the humane endpoint guideline, mice losing 20% of their body weight relative to the baseline weight were euthanized immediately after observation by isoflurane inhalation (euthanized mice, n = 156; found dead mice, n = 52). Mouse health and behavior were monitored daily, and temperature and humidity in

the cages were kept constant, and the bedding was changed once a week for optimal housing conditions.

## Viruses and cells

The seasonal H3N2 influenza virus vaccine strain A/Switzerland/9715293/2013 (SW293) was obtained from the World Health Organization (WHO) collaborating Center, National Institute of Infectious Diseases (NIID). The virus was subsequently inoculated into the allantoic cavities of specific pathogen-free (SPF) 10-day-old chicken eggs and cultured at 37˚C for 48 h. The allantoic fluid was harvested and filtered through a 0.2-μm syringe filter, and aliquots were stored at -80˚C until use. Madin-Darby canine kidney (MDCK) cells were purchased from the American Type Culture Collection (ATCC, Manassas, VA, USA) and maintained in Minimum Essential Medium Eagle (MEM, GenDEPOT, Katy, TX, USA) supplemented with 10% fetal bovine serum (FBS, Corning, New York, NY, USA).

## Mouse adaptation of seasonal H3N2 influenza virus

Viral adaptation was achieved by lung-(cell)-lung passaging, following a modified classical method [27]. Briefly, six-week-old female Balb/c or DBA/1J mice ($n$ = 3–7/group) (ORIENT) were anesthetized by intraperitoneal injection of avertin (200 mg/kg) before viral inoculation. Mice were intranasally inoculated with 50 μL phosphate-buffered saline (PBS) containing $1.4 \times 10^6$ plaque-forming units (PFU) of SW293, labeled as passage 0 or P0. At 3 days post-infection (dpi), the inoculated mice were euthanized to obtain lungs, and the lungs were homogenized in PBS. Supernatants were then collected by centrifugation and were labeled as the P1 lung homogenates. New mice were inoculated with 50 μL lung homogenate, and the procedure was repeated (lung-to-lung passage). An MDCK cell culture step was added between lung passages in case of lung-cell-lung passage. The mice ($n$ = 2–7/group) inoculated with $1.4 \times 10^6$ PFU SW293 or 50 μL lung homogenate obtained from infected mice for serial passage were observed daily to monitor disease signs and mortality rates for nine days.

## Plaque purification

To isolate single-phenotype viruses, plaque purification was performed with lung isolates of each passage using MDCK cells, as described previously [28]. Briefly, supernatants of lung homogenates were serially diluted 10-fold in PBS from $10^{-2}$ to $10^{-6}$. Confluent MDCK cells were prepared in 6-well plates and infected with 200 μL of the diluted samples. After 1 h, the cells were washed with PBS and overlaid with 2 mL 0.8% agarose–medium mixture containing 1 μg/mL L-1-tosylamide-2-phenylmethyl chloromethyl ketone (TPCK)-treated trypsin (Thermo Fisher Scientific, Waltham, MA, USA). Three to four days later, single-plaque colonies were selected from each plate, resuspended in medium, and propagated in MDCK cells. After 48–72 h of incubation, viruses were harvested, and the viral titer (PFU/mL) of each strain was calculated.

## DNA sequencing

Viral RNA from each sample was isolated using the QIAamp® Viral RNA Mini kit (Qiagen, Hilden, Germany) following the manufacturer's instructions. Reverse transcription of viral RNA was performed using the SuperScript III Reverse Transcriptase (Thermo Fisher Scientific). Subsequently, all gene segments were amplified using SuperTaq™ Plus Polymerase (Thermo Fisher Scientific). The PCR products were purified using the QIAquick® Gel Extraction kit (Qiagen). Sequencing was performed using a 3730 DNA analyzer (Applied Biosystems,

Foster City, CA, USA), and the sequencing results for all viruses were analyzed and aligned using the CLC Main Workbench software.

### Reassortant virus rescue

The eight gene segments of seasonal H3N2 IAV (A/Switzerland/9715293/2013; SW293) and five gene (PB2, PA, HA, NP, and NA) segments of the mouse-adapted H3N2 virus were amplified using reverse transcription-polymerase chain reaction and cloned into the plasmid vector pHW2000 provided by St. Jude Children's Research Hospital, Memphis. Reassortant H3N2 viruses, each containing one of the mutant genes from MA_SW, were rescued in the genetic background of SW using a reverse genetic system [29]. Briefly, MDCK and 293T cells were co-cultured in 6-well plates at 37˚C, and on the following day, a mixture of TransIT-LT1 (MIRUS) transfection reagent and 1 μg of each plasmid was added to the cells. After 24 h, Opti-MEM (Gibco, Gaithersburg, MD, USA) containing 0.5 μg/mL of TPCK-trypsin (Thermo Fisher Scientific) was added to the cells. Reassortant viruses were harvested from the supernatant of the cell culture from days 3 to 7 post-transfection and propagated in MDCK cells at 37˚C for three days. After analyzing the gene sequence of the reassortant viruses, they were titrated using the 50% tissue culture infectious dose ($TCID_{50}$) in MDCK cells using the Reed and Muench method [30]. The amplified viruses were stored at -80˚C for further studies.

### Mouse pathogenicity experiments

The 50% mouse lethal dose ($MLD_{50}$) of the SW293 and Balb/c-passaged clones was determined using Balb/c mice ($n$ = 4–5/group, ORIENT). Mice were intranasally inoculated with 50 μL of 10-fold serial dilution of each virus in PBS under general anesthesia using avertin (200 mg/kg). The number of surviving and dead mice were recorded for nine days. $MLD_{50}$ values were calculated by the Reed-Muench method after the observation period. To confirm pathogenicity of reassortant H3N2 viruses, Balb/c mice (ORIENT) were anesthetized by intraperitoneal injection of avertin (200 mg/kg) before viral inoculation. Mice ($n$ = 5/group) were intranasally inoculated with 30 μL of 10-fold serial diluted reassortant viruses ($10–10^5$ $TCID_{50}$/mL), followed by daily observation of weight changes and survival rates for 14 days to determine viral pathogenicity. The clinical signs of mice were monitored once a day during the experiment. Uninfected control mice were inoculated with the same volume of PBS. To determine the replication competence of reassortant viruses in mice, six mice from each group were inoculated intranasally with $10^5$ $TCID_{50}$ of the viruses ($10^3$ $TCID_{50}$ for the MA_SW virus), and nine organs (nasal turbinate, trachea, lung, brain, heart, spleen, liver, kidney, and small intestine) were collected from three mice per group at 3 and 6 dpi, respectively. All organs were homogenized with infection media containing TPCK-trypsin (Thermo Fisher Scientific), MEM vitamin (Gibco), and 1% penicillin-streptomycin (Gibco) in MEM (GenDEPOT), and the supernatant from the homogenized samples were collected by centrifugation. Viral titers of the supernatant were determined by $TCID_{50}$. The lung tissues of mice were fixed in 10% neutral formalin buffer for histologic analysis.

### Growth test of reassortant H3N2 viruses

To analyze the time-dependent growth profiles of reassortant H3N2 viruses, MDCK cells were infected at a multiplicity of infection (MOI) of 0.001 and were incubated at 37˚C. The supernatants were harvested at 3, 6, 12, 24, 48, and 72 h. The infectious viral titers were determined by $TCID_{50}$ in MDCK cells.

## Histopathology and immunohistochemical (IHC) analyses

Histopathologic examinations were conducted, as described previously [31]. Briefly, paraffin-embedded tissues were deparaffinized and hydrated with xylene and alcohols, followed by staining with hematoxylin and eosin and investigation of microscopic lesions in mouse lungs. For IHC analysis, lung tissues were incubated with an anti-IAV antibody (Merck, Kenilworth, NJ, USA) and a biotinylated secondary antibody (Dako; Agilent Technologies, Santa Clara, CA, USA) at 37˚C. After attaching with streptavidin alkaline phosphatase (Calbiochem, San Diego, CA, USA), treated tissues were colored using fast red dye (Sigma-Aldrich, St. Louis, MO, USA).

## Statistical analysis

All values are expressed as the mean of each cohort, and the error bar indicates the standard error of the mean (SEM). Student's $t$-test was used to compare between groups. A $p$-value less than 0.05 was considered statistically significant.

## Results

### Adaptation of seasonal H3N2 influenza virus to mice

To increase the virulence of seasonal H3N2 influenza in mice, SW293 was serially inoculated in a series of Balb/c or DBA/1J mice, and then the mouse mortality rates were monitored for nine days (Tables 1 and 2). Despite performing ten lung-cell-lung passages in Balb/c mice, animals were still not susceptible to the SW293 strain (Table 1). However, after three lung-cell-lung passages in DBA/1J mice, the virus was able to kill all mice in the group (Table 2). Thereafter, lung-to-lung passages were performed using DBA/1J mice until passage 13, and four clones each were acquired from passages 9, 11, 12, and 13 by plaque purification (D_P9_C1–4, D_P11_C1–4, D_P12_C1–4, D_P13_C1–4).

For Balb/c mice, adaptation was initiated by infecting DBA/1J_P12_C1, which was followed by ten lung-to-lung passages (Table 3). After five passages, all inoculated mice died; therefore, four clones each from passage 6, 8, and 9 were obtained (B_P6_C1–4, B_P8_C1–4, and B_P9_C1–4). Five mice per group were infected intranasally with $10^6$ PFU of Balb/c-passaged clones and mortality rates were monitored for nine days (Table 4). At least four out of the five

**Table 1. Balb/c mortality after serial SW293 infection.**

| Method | Passage | No. of dead mice / No. of total mice | | | Mortality (%) |
|---|---|---|---|---|---|
| | | Total dead mice | Found dead mice | Euthanized mice | |
| Lung-Cell-Lung | 0 | 0/3 | 0/3 | 0/3 | 0 |
| | 1 | 0/3 | 0/3 | 0/3 | 0 |
| | 2 | 0/3 | 0/3 | 0/3 | 0 |
| | 3 | 0/3 | 0/3 | 0/3 | 0 |
| | 4 | 0/3 | 0/3 | 0/3 | 0 |
| | 5 | 0/5 | 0/5 | 0/5 | 0 |
| | 6 | 0/3 | 0/3 | 0/3 | 0 |
| | 7 | 0/3 | 0/3 | 0/3 | 0 |
| | 8 | 0/6 | 0/6 | 0/6 | 0 |
| | 9 | 0/5 | 0/5 | 0/5 | 0 |
| | 10 | 0/5 | 0/5 | 0/5 | 0 |

Balb/c mice were inoculated with $1.4 \times 10^6$ PFU SW293 or 50 μL lung homogenate obtained from infected mice. Mortality rates were observed for nine days.

**Table 2. DBA/1J mortality after serial infection with SW293.**

| Method | Passage | No. of dead mice / No. of total mice | | | Mortality (%) |
|---|---|---|---|---|---|
| | | Total dead mice | Found dead mice | Euthanized mice | |
| Lung-Cell-Lung | 0 | 0/3 | 0/3 | 0/3 | 0 |
| | 1 | 0/2 | 0/2 | 0/2 | 0 |
| | 2 | 2/3 | 0/3 | 2/3 | 66.67 |
| | 3 | 5/5 | 0/5 | 5/5 | 100 |
| Lung-to-Lung | 4 | 5/5 | 0/5 | 5/5 | 100 |
| | 5 | 5/5 | 1/5 | 4/5 | 100 |
| | 6 | 5/5 | 0/5 | 5/5 | 100 |
| | 7 | 5/5 | 1/5 | 4/5 | 100 |
| | 8 | 5/5 | 1/5 | 4/5 | 100 |
| | 9 | 7/7 | 7/7 | 0/7 | 100 |
| | 10 | 7/7 | 6/7 | 1/7 | 100 |
| | 11 | 7/7 | 6/7 | 1/7 | 100 |
| | 12 | 7/7 | 0/7 | 7/7 | 100 |
| | 13 | 7/7 | 3/7 | 4/7 | 100 |

DBA/1J mice were inoculated with $1.4 \times 10^6$ PFU SW293 or 50 μL lung homogenate obtained from infected mice. Mortality rates were observed for nine days

clone-infected mice died except for B_P8_C1. In particular, all mice infected with B_P8_C2 died, and 80% of them died before reaching the euthanasia baseline (Table 4). In addition, the $MLD_{50}$ values of the Balb/c-passaged clones were about 1000-fold lower than that of the SW293 parental virus (S1 Table). These results demonstrate that the serial passage of SW293 parental virus results in markedly increased mortality and virulence in mice.

## Sequence analysis of mouse-adapted H3N2 viral genomes

To identify the mutations responsible for the increased ability to cause fatal infection, we initially sequenced the genomes of two Balb/c–and three DBA/1J–passaged clones (Table 5). A total of 20 amino acid substitutions involving five viral proteins were detected, which produced seven amino acid changes in HA, five in PA, three each in NP and NA, and two in PB2.

**Table 3. Balb/c mortality after serial infection with D_P12_C1.**

| Method | Passage | No. of dead mice / No. of total mice | | | Mortality (%) |
|---|---|---|---|---|---|
| | | Total dead mice | Found dead mice | Euthanized mice | |
| Lung-Cell-Lung | 0 | 2/5 | 0/5 | 2/5 | 40 |
| | 1 | 5/5 | 2/5 | 3/5 | 100 |
| | 2 | 4/5 | 0/5 | 4/5 | 80 |
| | 3 | 5/5 | 0/5 | 5/5 | 100 |
| | 4 | 3/5 | 0/5 | 3/5 | 60 |
| | 5 | 5/5 | 0/5 | 5/5 | 100 |
| | 6 | 5/5 | 4/5 | 1/5 | 100 |
| | 7 | 5/5 | 0/5 | 5/5 | 100 |
| | 8 | 5/5 | 4/5 | 1/5 | 100 |
| | 9 | 5/5 | 0/5 | 5/5 | 100 |
| | 10 | 4/4 | 0/4 | 4/4 | 100 |

Balb/c mice were inoculated with $1.8 \times 10^6$ PFU/mL D_P12_C1 or 50 μL lung homogenate obtained from infected mice. Mortality rates were observed for nine days.

**Table 4. Balb/c mortality after infection with Balb/c-passaged clones.**

| Virus | No. of dead mice / No. of total mice | | | Mortality (%) |
|---|---|---|---|---|
| | Total dead mice | Found dead mice | Euthanized mice | |
| B_P6_C1 | 5/5 | 0/5 | 5/5 | 100 |
| B_P6_C2 | 5/5 | 0/5 | 5/5 | 100 |
| B_P6_C3 | 4/5 | 0/5 | 4/5 | 80 |
| B_P6_C4 | 4/5 | 0/5 | 4/5 | 80 |
| B_P8_C1 | 1/5 | 0/5 | 1/5 | 20 |
| B_P8_C2 | 5/5 | 4/5 | 1/5 | 100 |
| B_P8_C3 | 5/5 | 0/5 | 5/5 | 100 |
| B_P8_C4 | 5/5 | 3/5 | 2/5 | 100 |
| B_P9_C1 | 5/5 | 0/5 | 5/5 | 100 |
| B_P9_C2 | 5/5 | 3/5 | 2/5 | 100 |
| B_P9_C3 | 5/5 | 0/5 | 5/5 | 100 |
| B_P9_C4 | 4/5 | 0/5 | 4/5 | 80 |

Balb/c mice were inoculated with $10^6$ PFU of Balb/c-passaged clones, and mortality rates were observed for nine days.

No mutations were detected in PB1, M1, M2, NS1, and NS2 genes. The five mouse-adapted viruses had all of the following substitutions: F332L in PB2, and A144T, T183A, F209S, and N262T in HA. Interestingly, there were five substitutions in only Balb/c-passaged clones, including V421F in PA, K18Q, and N160D in HA, D34N in NP, and S331R in NA. Amino acid positions are numbered based on translated amino acids of DNA sequences encoding SW293 proteins (EPI_ISL_166310).

To investigate whether the amino acid changes observed in mouse-adapted H3N2 viruses had been identified in other H3 influenza viruses, we matched mutations of the mouse-adapted viruses with the amino acids of the circulating H3N2 viruses at the same positions using the FluSurver database (http://flusurver.bii.a-star.edu.sg) (Table 6). We found that most H3 viruses possessed an H residue at PA position 713 (99.95%) and a T residue at HA position 144 (80.28%). In contrast, most identified substitutions in this study were rarely observed in natural isolates, accounting for less than 1% of all substitutions, with the exception of an S residue at HA position 209 (4.23%) and an R residue at NA position 331 (7.34%).

## Generation of reassortant H3N2 viruses carrying mutations derived from mouse adaptation

The B_P8_C2 strain named maSW293B8 showed significantly enhanced mortality and virulence in mice (Table 4 and S1 Table). This virus contained 13 amino acid substitutions

**Table 5. Amino acid substitutions identified in mouse-adapted H3N2 influenza viruses.**

| Virus | PB2 | | PA | | | | | HA | | | | | | | NP | | | NA | | |
|---|---|---|---|---|---|---|---|---|---|---|---|---|---|---|---|---|---|---|---|---|
| | 323 | 355 | 86 | 396 | 421 | 609 | 713 | 18 | 144 | 160 | 183 | 209 | 262 | 522 | 34 | 189 | 384 | 331 | 369 | 463 |
| SW_293 | F | R | M | E | V | K | Y | K | A | N | T | F | N | E | D | M | G | S | T | D |
| D_P9_C1 | L | R | M | E | V | K | Y | K | T | N | A | S | T | D | D | M | G | S | M | D |
| D_P11_C2 | L | R | I | E | V | R | H | K | T | N | A | S | T | E | D | M | R | S | A | N |
| D_P13_C1 | L | K | I | E | V | R | H | K | T | N | A | S | T | E | D | I | R | S | K | D |
| B_P8_C2 | L | R | I | E | F | R | H | Q | T | D | A | S | T | E | N | M | G | R | T | D |
| B_P9_C2 | L | R | I | K | F | R | H | Q | T | D | A | S | T | E | N | M | G | R | T | D |

**Table 6. Database search for amino acid substitutions observed in mouse-adapted H3N2 strains.**

| Gene | Position | Amino acid | Frequency (%) |
|------|----------|------------|---------------|
| PB2  | 323      | L          | -             |
|      | 355      | K          | 0.07          |
| PA   | 86       | I          | 0.05          |
|      | 369      | K          | 0.04          |
|      | 421      | F          | -             |
|      | 609      | R          | 0.02          |
|      | 713      | H          | 99.95         |
| HA   | 18       | Q          | -             |
|      | 144      | T          | 80.28         |
|      | 160      | D          | 0.01          |
|      | 183      | A          | 0.01          |
|      | 209      | S          | 4.23          |
|      | 262      | T          | 0.04          |
|      | 522      | D          | 0.02          |
| NP   | 34       | N          | 0.07          |
|      | 189      | I          | 0.58          |
|      | 384      | R          | 0.07          |
| NA   | 331      | R          | 7.34          |
|      | 369      | K          | 0.24          |
|      |          | A          | 0.06          |
|      | 463      | N          | 1.82          |

related to mouse adaptation of seasonal H3N2 virus, and five mutations (V421F in PA; K18Q and N160D in HA; D34N in NP; and S331R in NA) were detected only in Balb/c-passaged clones (Table 5). To confirm the contribution of these amino acid substitutions to pathogenicity, we generated four reassortant H3N2 viruses containing the mutations responsible for mouse adaptation using a reverse genetics system [29]. In the genetic background of the seasonal H3N2 virus, each mutant virus contained amino acid substitutions in either PB2, HA, NP, or NA and was named according to the gene carrying the mutation (Table 7). However, we failed to rescue the PA mutant virus, likely due to the loss of viral fitness induced by mutations in PA. Considering that complementary mutations to amino acid substitutions in PA occurred among other genes in RNP complex, we attempted to rescue the reassortant viruses that combined mutations in PA with those in other genes in the RNP complex, such as PB2+PA, PA+NP, and PB2+PA+NP; however, these strains could not be rescued for futher evaluation. A reassortant virus in which all genes comprise H3N2 was generated as seasonal IAV (RG_SW), and maSW293B8 was used as a mouse-adapted H3N2 virus (MA_SW).

**Table 7. Reassortant H3N2 viruses used in this study.**

| Virus | Protein | Mutation |
|-------|---------|----------|
| PB2 mutant | PB2 | F323L |
| PA mutant | PA | M86I, V421F, K609R, Y713H |
| HA mutant | HA | K18Q, A144T, N160D, T183A, F209S, N262T |
| NP mutant | NP | D34N |
| NA mutant | NA | S331R |

## Virulence of reassortant H3N2 viruses in mice

To confirm the morbidity and mortality associated with the reassortant H3N2 viruses, mice were infected with various doses of mutant viruses (10, $10^2$, $10^3$, $10^4$, or $10^5$ TCID$_{50}$) or PBS as a control, and weight changes and survival rates were monitored daily for two weeks. Additionally, we tested RG_SW and MA_SW using a similar protocol. In the case of MA_SW, higher doses were not used as $10^3$ TCID$_{50}$ showed sufficient lethality in mice. Mice injected with RG_SW, PB2 mutant, NP mutant, or NA mutant showed a ~10% weight loss but recovered within 4 dpi (Fig 1A, 1B, 1D and 1E). By contrast, infection with $10^5$ TCID$_{50}$ of the HA mutant resulted in a ~20% weight loss at 3 dpi (Fig 1C), and the bodyweight of the mice inoculated with MA_SW was reduced by 20%, except for one mouse (Fig 1F).

Changes in survival rate were similar to those observed with weight loss (Fig 2). All mice survived infection with RG_SW, PB2 mutant, NP mutant, or NA mutant; however, the mortality rate of mice infected with $10^5$ of the HA mutant was at 40% (Fig 2C). Additionally, following MA_SW infection, four of five mice died within 7 dpi (Fig 2F). These results demonstrated that mutations in the HA gene were the most crucial factor for enhanced virulence of mouse-adapted H3N2 virus.

## Viral replication in the mice infected with reassortant H3N2 viruses

To examine organ-specific replication of viruses containing mouse-adapted mutations, mice were infected intranasally with RG_SW ($10^5$ TCID$_{50}$), mutant viruses ($10^5$ TCID$_{50}$), or MA_SW ($10^3$ TCID$_{50}$), and organs were collected at 3 and 6 dpi. Organ-specific viral titers were measured using the TCID$_{50}$ assay (Fig 3). RG_SW and the NA mutant were re-isolated only in the nasal turbinate (NT) at 3 dpi and were not detected in any organs at 6 dpi (Fig 3B and 3F). The PB2 mutant was found only in the NT from two mice at 3 dpi, whereas the virus was isolated in the trachea and lung of one mouse at 6 dpi (Fig 3C). Additionally, the HA mutant, NP mutant, and MA_SW showed significantly higher viral titers in NT, trachea, lung, and heart relative to other viruses at 3 dpi (Fig 3D, 3E and 3G), although these viral titers

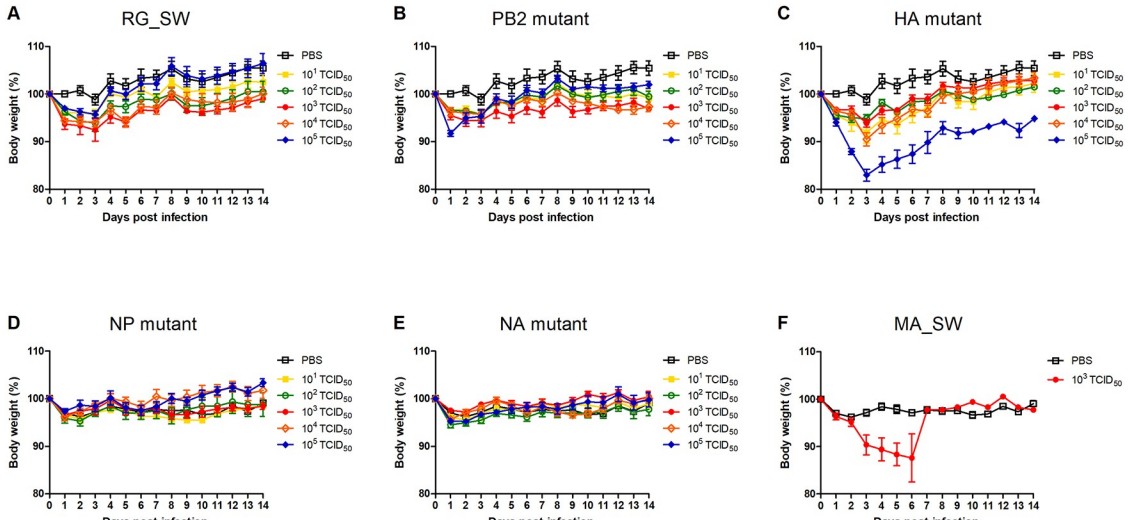

**Fig 1. Bodyweight changes of the mice infected with reassortant H3N2 viruses.** Five mice from each group were intranasally inoculated with reassortant H3N2 viruses (10–$10^5$ TCID$_{50}$), RG_SW (10–$10^5$ TCID$_{50}$), MA_SW ($10^3$ TCID$_{50}$), or mock-infected with PBS as a control. After inoculation, weight loss was monitored for two weeks in mice infected with (A) RG_SW, (B) PB2 mutant, (C) HA mutant, (D) NP mutant, (E) NA mutant, and (F) MA_SW.

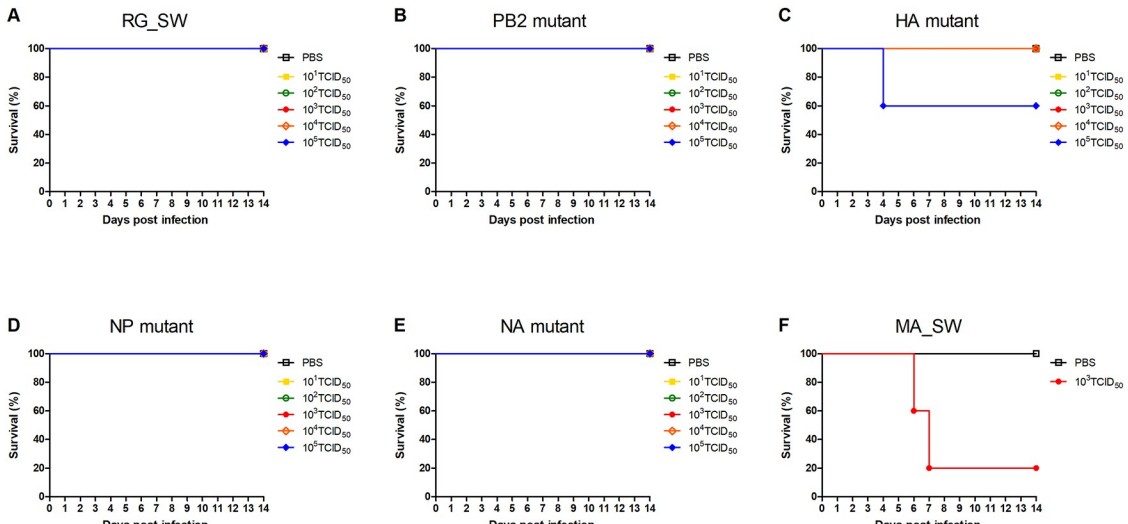

**Fig 2. Survival rates of mice infected with reassortant H3N2 viruses.** Groups of mice were intranasally inoculated with (A) RG_SW, (B) PB2 mutant, (C) HA mutant, (D) NP mutant, (E) NA mutant, or (F) MA_SW H3N2 viruses as described in Fig 1. Survival rates of the mice were observed daily for two weeks.

decreased at 6 dpi, with the reductions different depending on the virus. The viral titer of the HA mutant decreased slightly in NT at 6 dpi; however, the virus was not found in other organs (Fig 3D). NP mutations were detected in NT and lung at 6 dpi, but the titer was significantly decreased as compared to that at 3 dpi (Fig 3E). Furthermore, the MA_SW titer decreased by 50% in NT, trachea, and lung at day six relative to that at 3 dpi but maintained its level at $>10^2$ $TCID_{50}$ (Fig 3G). These data indicated that the amino acid substitutions in HA and NP were responsible for increased replication in mice. Notably, the NP mutant that did not show virulence in infected mice (Figs 1D and 2D) and replicated in mouse organs with similar efficiency as the HA mutant that caused morbidity and mortality (Fig 3D and 3E).

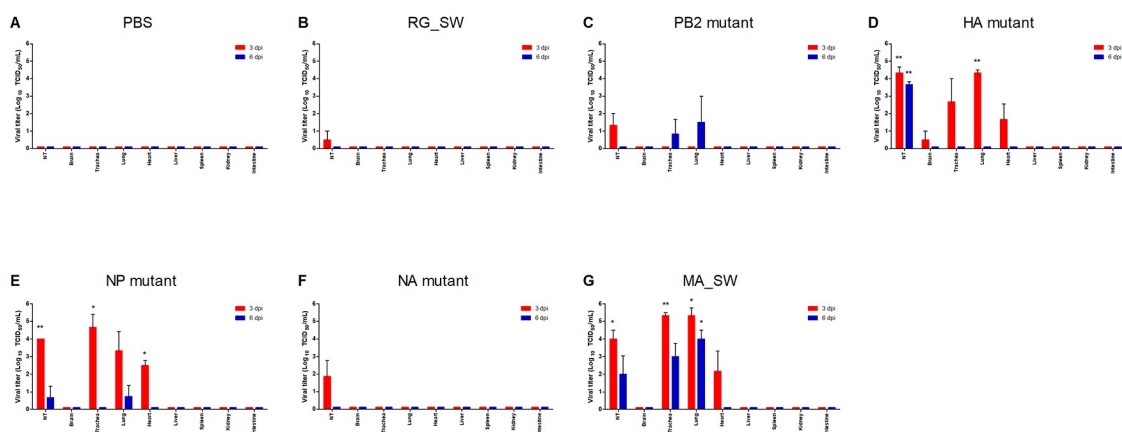

**Fig 3. Virus re-isolation from mice infected with reassortant H3N2 viruses.** Mice were intranasally inoculated with (A) 30 μL PBS (control), (B) $10^5$ $TCID_{50}$ RG_SW, (C) PB2 mutant, (D) HA mutant, (E) NP mutant, (F) NA mutant, or (G) $10^3$ $TCID_{50}$ MA_SW virus. Multiple organs (NT, trachea, lung, brain, heart, spleen, liver, kidney, and small intestine) were collected from three mice in each group at 3 and 6 dpi, and viral titers were determined according to the $TCID_{50}$ in MDCK cells. ($^*P < 0.05$, $^{**}P < 0.01$ compared with the PBS control group).

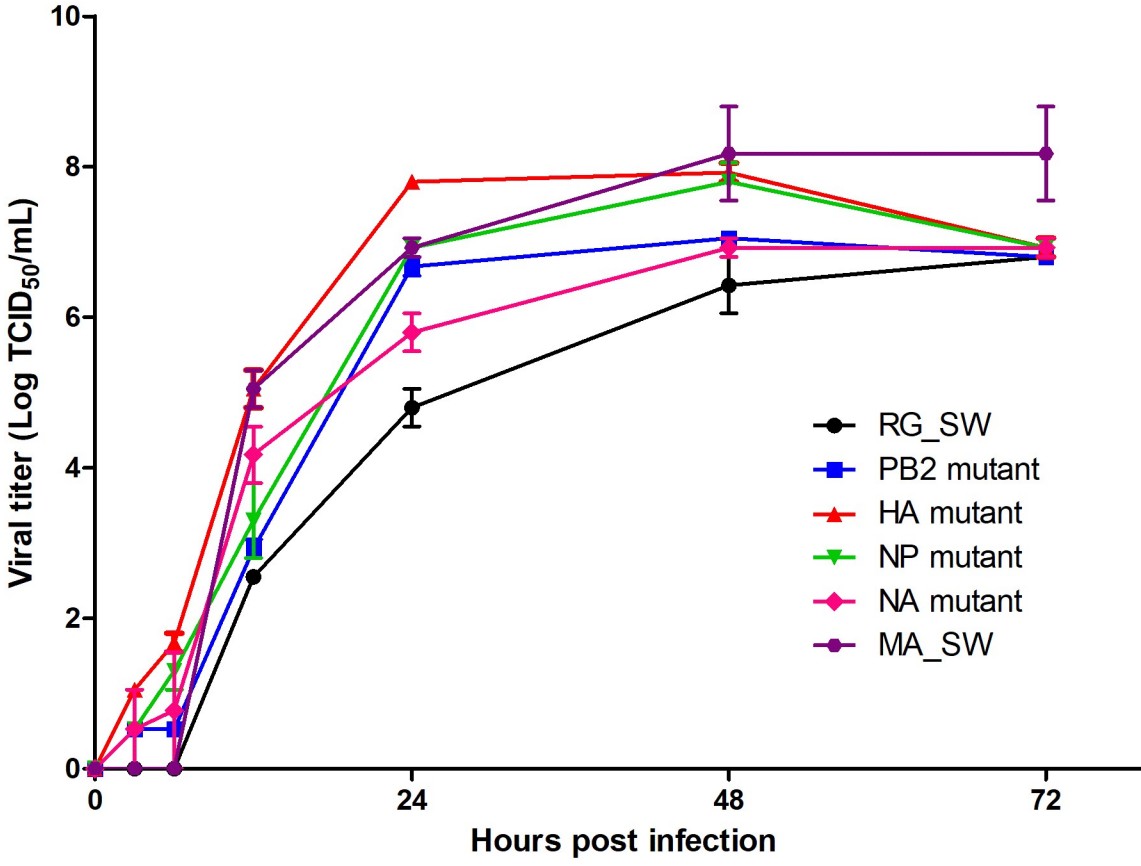

**Fig 4. Growth kinetics of reassortant H3N2 viruses.** MDCK cells were infected with 0.001 multiplicity of infection (MOI) of RG_SW, PB2 mutant, HA mutant, NP mutant, NA mutant, or MA_SW and were incubated at 37 °C. Supernatants were collected at 3, 6, 12, 24, 48, and 72 h after infection, and viral titers were measured by $TCID_{50}$ in MDCK cells.

### Growth characteristics of reassortant H3N2 viruses

Cell growth properties of reassortant H3N2 viruses were examined in MDCK cells at 3, 6, 12, 24, 48, and 72 h post-infection (hpi) (Fig 4). All reassortant viruses and MA_SW showed higher peak titer and faster growth rate than those of RG_SW. The HA mutant presented the highest peak titer at 24 hpi and the fastest growth rate among the reassortant H3N2 viruses. The initial growth rate of the NP mutant was moderate; however, it recorded the second-highest peak titer following the HA mutant. The viral titer of the PB2 mutant was similar to that of the NP mutant until 24 hpi, after which the titer was maintained. These results indicated that mouse-adapted mutations contribute to the increased replication ability in mammalian cells.

### Histopathological lesions and IHC analysis of lungs from mice infected with reassortant H3N2 viruses

Mice infected with MA_SW showed severe infiltration of round cells including interstitial macrophages and lymphocytes in peribronchial or perivascular regions, necrosis of the infiltrated round cells, mild hemorrhage, cell debris comprising necrotic cells and hemorrhagic red blood cells in the bronchial lumen, and thrombotic changes at 3 dpi and severe lung damage with loss of alveolar septa at 6 dpi (Fig 5A). Cellular debris in the bronchial lumen and moderate infiltration of round cells in the peribronchial or perivascular region were observed

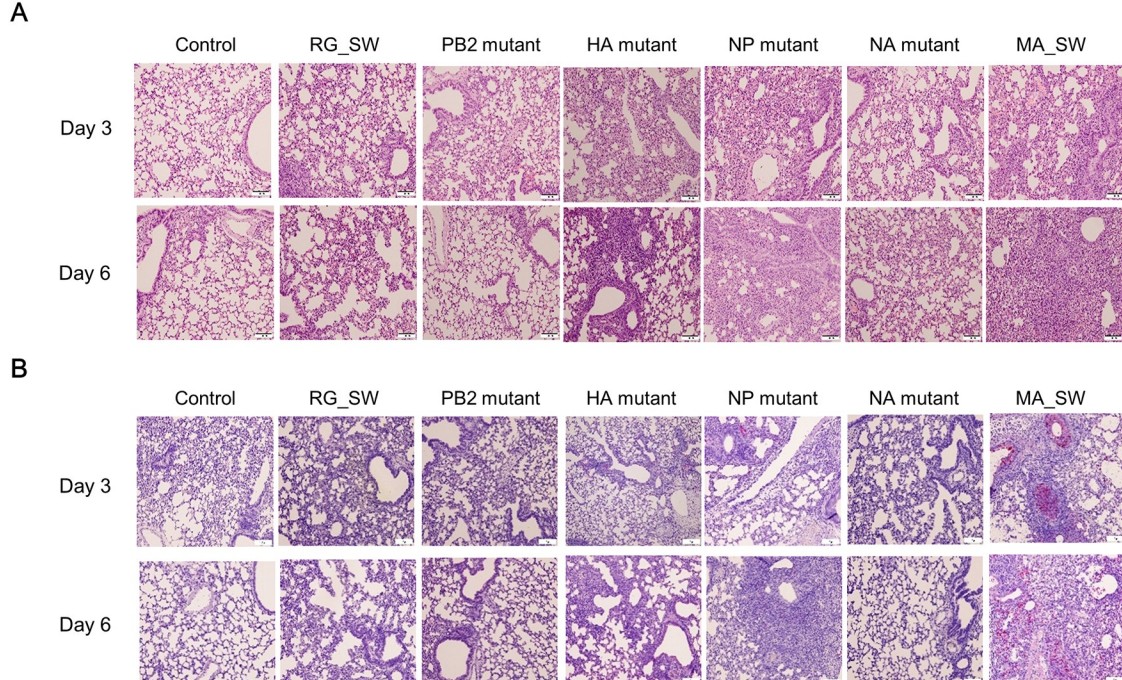

**Fig 5. Histopathological lesions and IHC analysis of mouse lungs infected with reassortant H3N2 viruses.** Lung tissues were collected from three mice in each group at 3 and 6 dpi for (A) histopathologic examination and (B) viral antigen detection by IHC analysis.

in mice infected with the HA mutant at 3 dpi, and mild hemorrhage in alveolar lumen was observed at 6 dpi. In mice infected with the NP mutant, diffuse alveolar damage was noted in addition to the most lesions occurred in mice infected with MA_SW. Mild thickening of the alveolar wall was the primary lesion found in mice infected with RG_SW and the NA mutant at 3 and 6 dpi, the PB2 and NP mutants at 3 dpi, and the NA mutant at 3 and 6 dpi. Mild peri-bronchial infiltration of interstitial macrophages was observed in the lungs of mice infected with the PB2 mutant at 6 dpi and the NP and NA mutants at 3 dpi. Moreover, hemorrhage in the alveolar lumen was observed in the NA-mutant-infected mice at 3 and 6 dpi. No lesions were found in uninfected control mice.

IHC staining showed similar results following virus re-isolation from mice infected with reassortant H3N2 viruses (Fig 5B). IAV antigens were detected in the lungs of mice infected with the HA mutant, NP mutant, and MA_SW and were mainly observed in bronchiolar epithelium and alveolar macrophages. However, no viral antigen was observed in the lung tissues infected with the PB2 or NA mutant.

## Discussion

Studies of molecular determinants of IAV pathogenicity and host adaptation are important to predict pre-pandemic viruses. Mice are the preferred animal model of IAV infection due to their relatively low cost, small size, and ease of handling [32]. Adaptation to mice is required for human-IAV infection of new hosts and acquisition of pathogenicity [33, 34], thereby making the mouse model suitable for analyzing the factors responsible for virulence and cross-species adaptability. We generated a mouse-adapted H3N2 virus [maSW293B8 (MA_SW)] derived from A/Switzerland/9715293/2013 by serial lung-cell-lung passages in Balb/c and

DBA/1J mice, which resulted in higher mortality rates in mice (Table 4 and S1 Table). Subsequently, we confirmed which mutations in MA_SW are major determinants of virulence and host adaptation in mice.

We rescued four reassortant viruses containing nine amino acid substitutions detected in the PB2, HA, NP, and NA genes of MA_SW except for the PA gene and evaluated their virulence, replication competence, and histopathology in mice. We found that the HA mutant was most similar to the mouse-adapted virus MA_SW in all respects as compared with other reassortant viruses. The mutant virus infected rapidly and replicated efficiently in multiple organs (Fig 3D) leading to significant weight loss and death in inoculated mice (Figs 1C and 2C). Additionally, the NP mutant infected and replicated equally efficiently as the HA mutant in mice (Fig 3D and 3E); however, its virulence was similar to that of the wild-type virus (Figs 1D and 2D). Moreover, the PB2 mutant did not cause rapid and widespread infection in mice but is considered to acquire mouse adaptability through enhanced replication competence (Figs 3C and 4). The three mutant viruses also showed higher peak titer and faster growth rate than that of the wild-type (Fig 4). These results suggested that mutations in the HA, NP, and PB2 genes are crucial factors for IAV pathogenicity and host range.

Six of the 13 mutations in MA_SW are harbored by the HA gene (Table 5), with four of these resulting amino acid substitutions (K18Q, N160D, T183A, and N262T) rarely detected in circulating seasonal H3N2 viruses (<0.05%) (Table 6). Notably, three substitutions (N160D, T183A, and N262T) might lead to loss of *N*-linked glycosylation of HA (Asn-X-Ser/Thr, where X is any amino acid, other than proline) [35]. The mutations related to deglycosylation of HA are often observed in mouse-adapted viruses, with many studies suggesting that these substitutions are responsible for increased virulence [7, 36–38]. Lack of glycosylation might influence HA-receptor-binding properties [38] or cleavage ability [7, 36]; however, mutations at N-linked glycosylation sites do not necessarily result in loss of glycosylation. Determination of whether these HA mutations lead to deglycosylation is necessary to confirm the association between deglycosylation and increased pathogenicity.

The D34N substitution in NP is rarely identified in circulating H3N2 viruses (0.07%) (Table 6); however, this mutation has been frequently detected in mouse-adapted H3N2 viruses according to previous studies [7, 9]. These results suggest that NP D34N is an important factor in mouse-adapted IAV. In the present study, the NP mutant containing the D34N mutation efficiently replicated in MDCK cells (Fig 4) and mouse organs (Fig 3E) without enhancing morbidity and mortality (Figs 1D and 2D). The exact role of the NP D34N mutation is not yet known but expected to be related to interactions with the RNP complex and viral polymerase activity [9, 39]. Analysis of the effects of the NP mutation on viral polymerase activity is required to determine how the NP D34N mutation contributes to enhanced replication competence in mice. Although the NP mutant showed low pathogenicity in mice (Figs 1D and 2D), it presented high replication efficiency in the mouse organs (Fig 3E) and MDCK cells (Fig 4), and severe histopathological lesions in the lung (Fig 5). In previous studies, virulence of the reassortant virus containing NP D34N was enhanced by increasing the dose inoculated into mice [9, 40]. Therefore, the pathogenicity of the NP mutant could be enhanced by increasing the dose. Taken together, NP D34N may be considered as a determinant of IAV pathogenicity.

In infected mice, the PB2 mutant was detected in only NT at 3 dpi but was re-isolated from the trachea and lung at 6 dpi (Fig 3C). These viral titers at day six were the highest among the mutant viruses (Fig 3) and indicated that this mutant is not as successfully infective as the HA and NP mutants but still replicates efficiently in mice. In MDCK cells, the initial growth rate of PB2 mutant was moderate until 6 hpi; however the mutant showed a rapid growth rate by 24 hpi (Fig 4). Mouse-adapted mutations in PB2 are major factors in determining IAV

pathogenicity and host adaptation [8, 9, 18] and involved in interactions with importin α1 and enhanced nuclear import of RNP [9, 18]. Moreover, the F323L mutation in PB2 may be associated with the high-yield properties of IAVs in MDCK cells [41]. These results suggest that the increased replication competence of the PB2 mutant might be due to the amino acid substitution responsible for viral polymerase activity.

The most severe microscopic lesions in the lung were observed in mice infected with MA_SW (Fig 5), which caused 80% mortality (Fig 2F). Moreover, the NP mutant exhibiting no lethality produced lung lesions with a higher degree of severity than those associated with the HA mutant, which produced 40% mortality (Fig 2C and 2D). Furthermore, the viral titer and re-isolated viruses from organs at 3 and 6 dpi did not differ between the HA and NP mutants (Fig 3D and 3E). Additionally, the NP mutant showed a slower initial growth rate and lower peak titer than that of the HA mutant in MDCK cells (Fig 4). HA plays a key role in IAV virulence [42], with cytokine production closely related to host tissue damage and fatality following IAV infection [43]. In the present study, it was unclear why the NP mutant produced less mortality but more severe histopathological lesions in the lung relative to the HA mutant (Fig 5A). Future studies are required to elucidate the virulence factors associated with the HA mutant.

As noted, we confirmed that the HA mutant was most similar to the mouse-adapted virus in all respects and relative to other reassortant viruses, suggesting that changes in the HA gene are major determinants of pathogenicity and host adaptation. However, the morbidity, mortality, and replication rate associated with HA mutant infection were lower relative to those of mouse-adapted viruses (Figs 1C, 1F, 2C, 2F, 3D and 3G) except for initial growth rate in MDCK cells (Fig 4). These results suggested that mutations in genes other than HA also contribute to mouse adaptation of seasonal H3N2 IAV. As follow-up studies, we plan to design reassortant viruses containing mouse-adapted mutations in multiple genes in order to analyze the characteristics of these viruses in mice. These studies will improve our understanding of the interactions and synergy between mouse-adapted genes.

Our study was conducted at the genetic level (Table 7). Characteristics of the HA mutant are comprehensive results of six amino acid substitutions in HA. Thus, these results are limited in explaining the effect of each amino acid substitution on pathogenicity and host adaptation of influenza virus. To identify the precise cause of the increased virulence of HA mutations, studies to analyze pathogenicity with recombinant viruses containing single point mutations need to be supplemented.

Overall, this study generated mouse-adapted H3N2 IAV and confirmed the contribution of each gene derived from the virus to increasing its pathogenicity and host adaptation. Further studies are required to analyze which mouse-adapted mutations cause enhanced virulence and host adaptation, as well as their underlying mechanisms. Our findings show that amino acid substitutions in HA and NP are crucial genetic markers responsible for viral pathogenicity and cross-species adaptability. Moreover, a genetic change in PB2 supports a novel host adaptation pathway through enhanced virus replication. Overall, this study provides new insights for understanding the molecular basis of acquired IAV pathogenicity and host adaptation and suggests that mouse-adapted changes in HA, NP, and PB2 might represent potential indicators of future pandemic IAVs.

## Supporting information

**S1 Table. MLD$_{50}$ of Balb/c-passaged clones in Balb/c mice.** Balb/c mice were inoculated with SW293, B_P8_C2, or B_P9_C2, and the live and dead mice were counted daily for nine days.

The 50% mouse lethal dose (MLD$_{50}$) was calculated using Reed-Muench method.
(PDF)

## Acknowledgments

We are grateful to Prof. Robert G. Webster at St. Jude Children's Research Hospital, at Memphis, for the plasmid vector pHW2000.

## Author Contributions

**Conceptualization:** Myung Guk Han.

**Data curation:** Eun-Ji Choi, Young Jae Lee, Byeongwoo Ahn, Myung Guk Han.

**Formal analysis:** Eun-Ji Choi, Young Jae Lee, Byeongwoo Ahn.

**Funding acquisition:** Myung Guk Han.

**Investigation:** Eun-Ji Choi, Young Jae Lee, Jin-Moo Lee, Yeon-Jung Kim.

**Project administration:** Eun-Ji Choi.

**Resources:** Eun-Ji Choi, Jin-Moo Lee, Yeon-Jung Kim.

**Supervision:** Kisoon Kim, Myung Guk Han.

**Validation:** Eun-Ji Choi, Young Jae Lee.

**Visualization:** Eun-Ji Choi, Young Jae Lee.

**Writing – original draft:** Young Jae Lee.

**Writing – review & editing:** Jang-Hoon Choi, Myung Guk Han.

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
