## [Decision Letter · Decision Letter 0]

4 Oct 2019

PONE-D-19-21391

The effect of mutations derived from mouse-adapted H3N2 seasonal influenza A virus to pathogenicity and host adaptation

PLOS ONE

Dear Dr. Han,

Thank you for submitting your manuscript to PLOS ONE. After careful consideration, we feel that it has merit but does not fully meet PLOS ONE’s publication criteria as it currently stands. Therefore, we invite you to submit a revised version of the manuscript that addresses the points raised during the review process.

We would appreciate receiving your revised manuscript by Nov 18 2019 11:59PM. To enhance the reproducibility of your results, we recommend that if applicable you deposit your laboratory protocols in protocols.io, where a protocol can be assigned its own identifier (DOI) such that it can be cited independently in the future. For instructions see: http://journals.plos.org/plosone/s/submission-guidelines#loc-laboratory-protocols

We look forward to receiving your revised manuscript.

Kind regards,

Man-Seong Park, Ph.D.

Academic Editor

PLOS ONE

**Journal Requirements:**

2. To comply with PLOS ONE submissions requirements, please provide method(s) of euthanasia in the Methods section of your manuscript.

3. In your Methods section, please give the sources of any cell lines used in your study.

4. Please note that all PLOS journals ask authors to adhere to our policies for sharing of data and materials: https://journals.plos.org/plosone/s/data-availability. According to PLOS ONE’s Data Availability policy, we require that the minimal dataset underlying results reported in the submission must be made immediately and freely available at the time of publication. As such, please remove any instances of 'unpublished data' or 'data not shown' in your manuscript and replace these with either the relevant data (in the form of additional figures, tables or descriptive text, as appropriate), a citation to where the data can be found, or remove altogether any statements supported by data not presented in the manuscript.

**Comments to the Author**

1. Is the manuscript technically sound, and do the data support the conclusions?

Reviewer #1: Partly

Reviewer #2: Yes

2. Has the statistical analysis been performed appropriately and rigorously? 

Reviewer #1: Yes

Reviewer #2: Yes

3. Have the authors made all data underlying the findings in their manuscript fully available?

Reviewer #1: Yes

Reviewer #2: Yes

4. Is the manuscript presented in an intelligible fashion and written in standard English?

Reviewer #1: Yes

Reviewer #2: Yes

5. Review Comments to the Author

Reviewer #1: The authors characterize mutations derived from mouse adaptation of an H3N2 seasonal influenza virus strain.

In many places of the manuscript, the description is ambiguous and logically convoluted. There are also many blunders. The work is interesting, but experimental procedures and results need to be described more clearly and logically.

Major points:

1. A figure showing increase of viral titer in mice as the number of passage increases is recommended.

2. Was each mutation from each individual colony or all from a single colony? How many colonies were picked? In how many colonies was each mutation found? This information may be added to Table 1. If all from a single colony, different combinations of mutations in addition to single mutations should have been studied. Description of the process of identifying the mutations is recommended.

3. Growth curves of the subject viruses (mutants and MA_SW) in MDCK cells and A549 cells are recommended. This is to see if the mutations were simple growth adaptations or host adaptations. The authors mention viral growths in MDCK cells in many places of the manuscript without actually showing the data.

4. Since the input titers of the mutant viruses and MA_SW are different, the input titers of the mutants being 100-fold higher than MA_SW, it is difficult to determine the connection between each mutation and the MA characteristics. According to Fig 1 and 2, the mutants didn’t show MA characteristics at the titer where MA_SW was infected. Fig 3 and 4 cannot be interpreted properly for the same reason. To the reviewer, it appears that none of the mutations individually have notable MA characteristics. The authors didn’t make any effort to figure out which combination of the mutations recapitulates the MA_SW phenotype.

5. Since the virus of PA mutation alone was not rescuable, at least a combination of PA with other mutations could have been tried. Since the D34N of NP is frequently found in MA H3N2, combinations of the NP mutation with PA and PB2 mutations could have been tested. If all mutations were found in the same colony (there is no information about this in current manuscript!), please do the mutation combination experiments.

Minor points:

1. Is the amino acid position numbering a CDS numbering (not clear especially for HA)?

2. Line 146, title of Table 1 is wrong. Reorganizing the mutations by passage stage (i.e., passage in DBA/1J or Balb/c) is recommended.

3. Lines 152 – 157, please try to write more clearly.

4. Lines 168 – 169, description doesn’t agree with the data.

5. Lines 239 – 241, where is the reference?

6. Line 243 – 245, check ‘nine amino acid substitutions’ and ‘seroconversion’ if they are correct.

7. Lines 275 – 279, the authors appear to be mixed up about mouse and human.

Reviewer #2: This study analyzed genetic markers for virulence and adaptation of a mouse-adapted seasonal influenza H3N2 strains. Four reassortant viruses were generated through reverse genetics and were evaluated for their morbidity and mortality in a mouse model. The data suggested that HA mutations were the most pronounced markers for enhanced virulence. The claims are supported by the data, successfully determining the genetic markers for mouse adaptation and increased virulence of a human-infecting IAV. Overall, this study seems suitable for publication to the journal, but some minor revisions are suggested.

1. It would be great if passage numbers of appearance of mutations are presented. The order of mutations of each gene, especially HA, will provide a valuable insight into tracking mutations during the host adaptation.

2. Lines 275~279: The authors stated that NP mutant virus was safe and the mutation might be a strategy to attenuate the virus. However, Figure 3 and Figure 4 show that NP mutant virus has increased ability to replicate in several organs as compared to the parent virus. Also, given that the NP mutation was induced by adaptation process in mice, the mutation is likely to be associated with increased virulence. The statement should be rephrased.

3. The time-dependent growth kinetics of each mutant virus on a cell culture (such as MDCK cell) would provide a better interpretation on replication abilities of the viruses. Especially, it would be interesting to examine whether delayed replication of the PB2 mutant in vivo (Fig 3) can be reproduced in vitro cell culture.

4. Line: 152~154: What does this sentence mean "We found that the MA_SW titer was lower than that of other viruses (3.16×104 TCID50/mL)…? Does it mean that the virus did not replicate to high titer enough in cell culture? Please clarify the meaning of this sentence.

6. PLOS authors have the option to publish the peer review history of their article (what does this mean?). If published, this will include your full peer review and any attached files.

Reviewer #1: No

Reviewer #2: No

---

## [Author Response · Author response to Decision Letter 0]

12 Nov 2019

I would like to thank the editorial board and the reviewers for reviewing the manuscript and providing constructive comments. We have extensively revised the manuscript based on the comments from the reviewers and have also included additional figures and tables in the revised manuscript. I believe that the revision has improved the quality of the manuscript and that it is not suitable for publication in your journal. We have also included details of the funding agency and grant that supported the study in the revised manuscript. A separate file with a point-by-point response to the reviewers’ comments has also been provided.

---

## [Editor Report · Decision Letter 1]

20 Dec 2019

The effect of mutations derived from mouse-adapted H3N2 seasonal influenza A virus to pathogenicity and host adaptation

PONE-D-19-21391R1

Dear Dr. Han,

We are pleased to inform you that your manuscript has been judged scientifically suitable for publication and will be formally accepted for publication once it complies with all outstanding technical requirements.

With kind regards,

Man-Seong Park, Ph.D.

Academic Editor

PLOS ONE
---

## [Editor Report · Acceptance letter]

26 Dec 2019

PONE-D-19-21391R1 

The effect of mutations derived from mouse-adapted H3N2 seasonal influenza A virus to pathogenicity and host adaptation 

Dear Dr. Han:

I am pleased to inform you that your manuscript has been deemed suitable for publication in PLOS ONE. Congratulations! Your manuscript is now with our production department. 

With kind regards,

on behalf of

Professor Man-Seong Park 

Academic Editor

PLOS ONE